# Electromagnetic Transduction Therapy (EMTT) Enhances Tenocyte Regenerative Potential: Evidence for Senolytic-like Effects and Matrix Remodeling

**DOI:** 10.3390/ijms26157122

**Published:** 2025-07-24

**Authors:** Matteo Mancini, Mario Vetrano, Alice Traversa, Carlo Cauli, Simona Ceccarelli, Florence Malisan, Maria Chiara Vulpiani, Nicola Maffulli, Cinzia Marchese, Vincenzo Visco, Danilo Ranieri

**Affiliations:** 1Trinity Centre for Biomedical Engineering, Trinity College, D02 R590 Dublin, Ireland; mancinim@tcd.ie; 2Department of Medical-Surgical Sciences and Translational Medicine, Sapienza University of Rome, 00189 Rome, Italy; mario.vetrano@uniroma1.it (M.V.); mariachiara.vulpiani@uniroma1.it (M.C.V.); n.maffulli@qmul.ac.uk (N.M.); 3Department of Life Sciences, Health and Health Professions, Link Campus University, 00165 Rome, Italy; a.traversa@unilink.it (A.T.); d.ranieri@unilink.it (D.R.); 4Department of Clinical Internal, Anaesthesiologic and Cardiovascular Sciences, Sapienza University of Rome, 00161 Rome, Italy; carlo.cauli@uniroma1.it; 5Department of Experimental Medicine, Sapienza University of Rome, 00161 Rome, Italy; simona.ceccarelli@uniroma1.it (S.C.); cinzia.marchese@uniroma1.it (C.M.); 6Department of Biomedicine and Prevention, Laboratory of Signal Transduction, Tor Vergata University of Rome, 00133 Rome, Italy; malisan@med.uniroma2.it; 7Department of Clinical and Molecular Medicine, Sapienza University of Rome, 00189 Rome, Italy

**Keywords:** EMTT, human tenocytes, tendinopathy, mechanotransduction, senescence, senolytic, tendon regeneration, extracellular matrix remodeling, cell migration

## Abstract

Tendinopathies are a significant challenge in musculoskeletal medicine, with current treatments showing variable efficacy. Electromagnetic transduction therapy (EMTT) has emerged as a promising therapeutic approach, but its biological effects on tendon cells remain largely unexplored. Here, we investigated the effects of EMTT on primary cultured human tenocytes’ behavior and functions in vitro, focusing on cellular responses, senescence-related pathways, and molecular mechanisms. Primary cultures of human tenocytes were established from semitendinosus tendon biopsies of patients undergoing anterior cruciate ligament (ACL) reconstruction (*n* = 6, males aged 17–37 years). Cells were exposed to EMTT at different intensities (40 and 80 mT) and impulse numbers (1000–10,500). Cell viability (MTT assay), proliferation (Ki67), senescence markers (CDKN2a/INK4a), migration (scratch test), cytoskeleton organization (immunofluorescence), and gene expression (RT-PCR) were analyzed. A 40 mT exposure elicited minimal effects, whereas 80 mT treatments induced significant cellular responses. Repeated 80 mT exposure demonstrated a dual effect: despite a moderate decrease in overall cell vitality, increased Ki67 expression (+7%, *p* ≤ 0.05) and significant downregulation of senescence marker CDKN2a/INK4a were observed, suggesting potential senolytic-like activity. EMTT significantly enhanced cell migration (*p* < 0.001) and triggered cytoskeletal remodeling, with amplified stress fiber formation and paxillin redistribution. Molecular analysis revealed upregulation of tenogenic markers (Scleraxis, Tenomodulin) and enhanced Collagen I and III expressions, particularly with treatments at 80 mT, indicating improved matrix remodeling capacity. EMTT significantly promotes tenocyte proliferation, migration, and matrix production, while simultaneously exhibiting senolytic-like effects through downregulation of senescence-associated markers. These results support EMTT as a promising therapeutic approach for the management of tendinopathies through multiple regenerative mechanisms, though further studies are needed to validate these effects in vivo.

## 1. Introduction

Acute and chronic tendinopathies are a major musculoskeletal health challenge in modern medicine, with a substantial socioeconomic impact on healthcare systems worldwide [1]. The burden of these conditions extends beyond direct medical costs to include reduced productivity, decreased quality of life, and long-term disability [2]. To address these challenges, the international scientific community has recently established consensus statements emphasizing the need for standardized approaches in diagnosing and treating these conditions, highlighting the complexity of tendon pathologies [3,4]. The pathophysiology of tendon disorders involves intricate inflammatory [5] and failed healing response mechanisms and complex cellular interactions that contribute to tissue dysfunction [6]. Multiple factors contribute to disease progression, including tissue degeneration, altered cellular responses, dysfunctional matrix organization, and significantly impaired healing capacity. In this challenging environment, traditional treatments often are ineffective [7,8].

Several innovative clinical approaches have recently emerged in the quest for effective treatments. Biological interventions, particularly platelet-rich plasma (PRP), have gained considerable attention given their potential to enhance tendon healing by releasing growth factors and bioactive molecules. However, their variable efficacy suggests the need for more refined treatment protocols and better patient selection criteria [9]. Similarly, mesenchymal stem cell (MSCs)-based therapies have emerged as promising strategies, offering the potential to restore tissue function through their differentiation capacity and paracrine effects [10]. Among physical therapy modalities, extracorporeal shock wave therapy (ESWT) has established itself as an effective treatment option for various musculoskeletal conditions [11]. In vitro studies have demonstrated the ability of ESWT to enhance the functional activities of tenocytes, including cell proliferation, migration, and matrix production, providing a scientific rationale for its clinical application [12,13,14].

Considering the principles of mechanotransduction and the success of ESWT [12], electromagnetic transduction therapy (EMTT) has recently emerged as an innovative therapeutic approach. This technology is a non-invasive regenerative modality that delivers high-frequency pulsed electromagnetic fields to stimulate cellular activity and promote tissue repair, representing a significant advancement in regenerative medicine and offering unique advantages in the treatment of musculoskeletal disorders [15,16]. Initial clinical trials have shown promising results in low back pain, demonstrating safety and efficacy [17]. Furthermore, studies on Achilles tendinopathy have reported significant improvements in patient outcomes [15]. The therapeutic potential of EMTT extends to other common tendinopathies, with documented success in managing lateral epicondylitis [18]. Particularly noteworthy are the positive results observed in treating rotator cuff disorders, where traditional therapies often show limited success [19]. Clinical investigations have consistently reported significant improvements in pain reduction and functional recovery in patients with chronic conditions [20]. Moreover, emerging evidence suggests enhanced therapeutic outcomes when EMTT is combined with other treatment modalities, particularly shockwave therapy, indicating potential synergistic effects in tissue healing and regeneration [19].

Despite the growing clinical evidence supporting EMTT’s therapeutic potential, the underlying biological mechanisms driving its effects remain a matter of debate. The complexity of cellular responses to mechanical and electromagnetic stimulation indicates multiple potential pathways through which EMTT could influence tissue healing and regeneration [21]. Understanding these mechanisms is crucial for several reasons: it may enhance treatment protocols, redefine patient selection criteria, and potentially lead to new therapeutic applications. The interaction between electromagnetic fields and cellular behavior represents a fascinating area of research that could revolutionize our approach to tissue regeneration and repair.

The present study aims to investigate the effects of EMTT on primary cultured human tenocyte behavior and function. By examining key cellular processes, including proliferation, migration, and matrix production, we seek to better elucidate the molecular and cellular mechanisms that may explain the observed clinical benefits of this promising therapeutic approach.

## 2. Results

### 2.1. Primary Cultures Characterization

Cellular morphology was evaluated using phase contrast microscopy. All primary cultured human tenocytes displayed a typical fibroblast-like phenotype with no detectable differences among cultures. Moreover, to immunocytochemically characterize tenocyte cultures, cells were stained with 4′,6-diamidino-2-phenylindole (DAPI), a commonly used stain to identify nuclei, and vimentin, a mesenchymal cell marker that labels cytoskeleton intermediate filaments. Additionally, pan-cytokeratin (pCK), which is a classical epithelial marker, was used as a negative control. Positive staining for perinuclear cytoplasmic bundles of filaments and the absence of pan-cytokeratin signal confirmed unequivocally the mesenchymal origin of all our cultures and excluded possible derangement of the cells (Figure 1).

### 2.2. Experimental Design

To evaluate the effects of EMTT on our cell cultures, our study was designed as a dose–response investigation, particularly given the lack of previous studies employing the same exposure model. To this end, two parameters were modulated during cell stimulation: magnetic field intensity and magnetic pulse frequency. The initial part of the study aimed to determine whether the magnetic field intensity would have an impact on cell viability. Using the parameters from a previous clinical study as a starting point [15,19], we decided to stimulate human primary tenocytes with two field intensities, namely 40 and 80 mT, while keeping the number of pulses constant at 1000 (Figure 2a). In the second part of the study, we aimed to amplify the cellular response by using the higher field intensity (80 mT) and increasing the number of impulses to 3500, which were then delivered up to three times in a row (Figure 2b).

### 2.3. Effects of EMTT on Cell Viability and Proliferation

Since there is no current literature on EMTT and its in vitro effect on tenocytes, we performed a preliminary experiment with a single exposure of 40 mT for 1000 impulses to assess its impact on cell viability and proliferation. No differences in cell vitality assessed by MTT between the controls and treated samples were observed at this intensity. EMTT-mediated proliferative effect was evaluated by a quantitative analysis of the percentage of Ki67+ cells. Immunofluorescence analysis also showed no differences in the single exposure experiment at 40 mT. After this initial assessment, to amplify the cellular response, we performed a more intense treatment consisting of 1000 impulses at 80 mT, the maximum setting supported by the MAGNETOLITH device. Subsequently, we also tested treatments of 3500 impulses at 80 mT, administered to three separate samples, repeated consecutively one, two, or three times. These higher intensity protocols significantly decreased cell vitality. To highlight differences in cell proliferation, the treated samples exhibited a 7% upregulation in Ki67+ cells. Quantitative evaluation of Ki67+ cells clearly showed significant variation in the proliferation marker in triple EMTT treatment (*p* ≤ 0.05) (Figure 3).

### 2.4. Molecular Analysis of EMTT-Exposed Tenocytes

RT-PCR was used to assess the effects of EMTT on human primary cultures to evaluate possible changes in the expression of typical tenocytes markers, such as Scleraxis (SCX), Tenomodulin (TNMD), Tenascin-C (TNC) [22], type I and III Collagens (COL1A1-COL3A1) [23], and also a differentiation marker for activated tenocytes such as α-SMA (ACTA2) [22] or a marker usually upregulated in senescence pathway such as p16 (CDKN2A) [24,25,26] (Figure 4a–h). Twenty-four hours after exposure, the tenocyte differentiation marker Scx was significantly enhanced. This was demonstrated in all samples except for the 80 mT, 3500 impulses, which were repeated three times in a row and followed a similar (not statistically significant) trend. Tenomodulin followed a similar upregulation pattern as SCX at 40 and 80 mT 1000 impulses, although only the 40 mT was statistically significant. Unexpectedly, in the repeated exposure to 80 mT 3500 impulses at 1×, 2×, and 3×, only the first treatment setting produced a significant increase, while the other settings showed downregulation, even though this was not statistically significant. Nevertheless, the upregulation of tenomodulin should be expected, since its gene expression is regulated by SCX [22]. Tenascin-C exhibited a positive trend across all treated samples, although statistical significance was reached only in cultures subjected to two or three repeated exposures. Cultures exposed to a single round of 3500 impulses showed a non-statistically significant negative trend.

Intriguingly, significant upregulation in the expression of Collagen type I and III was observed, both after a preliminary single dose treatment (40 or 80 mT for 1000 impulses) and a repeated, more intense treatment (80 mT for 3500 impulses 2×, 3×). In previous works on tendon regenerative therapies, Collagen I and III production was generally associated with wound healing [10,17,20]. Although α-SMA showed a downmodulation pattern in almost all treated samples, it was statistically significant only for 1 × 3500 impulses. CDKN2A was significantly downregulated in all treated samples compared to controls. Our results indicate that EMTT affects tenocyte differentiation and collagen production, both essential in tendon injury repair. In addition, to evaluate the expression of proteases, which can facilitate the breakdown of extracellular matrix (ECM) proteins and promote cellular migration, we focused on matrix metalloproteinases MMP9 [27,28]. Our results showed that a progressive enhancement in treatment intensity (80 mT for 3500 impulses, administered twice or three times) induces a significant upregulation in MMP9 expression, while treatment at 40 and 80 mT for 1k had no effect. Taken together, our results indicate that human primary cultured tendon-derived cells exposed to EMTT are affected by the treatments in different ways. This is demonstrated by variations in collagen production, metalloproteinase expression, and changes in tenocyte differentiation markers, which are crucial in the mechanisms guiding injury repair. The observed downregulation of CDKN2a suggests potential effects on senescence pathways, though definitive characterization of senolytic activity would require additional validated approaches such as senescence-associated β-galactosidase assays or single-cell analyses.

### 2.5. Scratch Test of EMTT-Treated Tenocytes

We performed a scratch test of cellular migration to evaluate the effect of our treatment because this assay can efficiently mimic a possible EMTT-induced in vitro wound healing. Cell migration was quantified by measuring the mean of % open residual areas after scratching, as described in Materials and Methods. For the migration test, we focused on the treatment that had the greatest impact on gene expression, especially in terms of metalloprotease expression; consequently, an alteration in biological behavior was expected, especially regarding cell migration. Quantitative analysis shows that the cell motility of EMTT-treated cells (80 mT for 3500 impulses 3×) was significantly increased (Figure 5). Overall, these findings suggest that cells exposed to EMTT exhibit enhanced cell migration properties, thereby promoting their wound-healing activity.

### 2.6. Actin Cytoskeleton Reorganization and Paxillin Localization

Since lamellipodia formation and membrane ruffles are hallmarks of cell migration [29], for the migration test, we analyzed the actin cytoskeleton organization of tendon-derived cells after EMTT exposure (80 mT for 3500 impulses 3×). The actin cytoskeleton organization revealed the presence of stress fibers, which are higher-order cytoskeletal structures composed of cross-linked actin filament bundles and myosin motor proteins, which enable cell contractility and cell motility. Cells, stained with TRITC–phalloidin to reveal filamentous actin cytoskeleton, showed defined stress fibres, whereas, in EMTT-treated cells, a significant increase in the number of stress fibres—thickly organized and spread throughout the cellular structure—was observed (*p* < 0.05). Thus, an evident migratory phenotype of the cells was achieved with EMTT treatment.

During migration, after the organization of protrusions, cells anchor the membrane to the substrate by the formation of new focal complexes. Paxillin, a component of the focal adhesions [30], is among the proteins found in these complexes [31]. Since this protein is expressed at the leading edge of the cells during the initiation of migration [32], a significant increase in the intracellular distribution of paxillin in EMTT-treated cells was observed (*p* < 0.01). Immunofluorescence analysis revealed that paxillin was uniformly distributed along each cell surface. However, in treated cells, it was specifically localized within focal adhesions, exhibiting pronounced migratory characteristics (Figure 6).

## 3. Discussion

Tendinopathies represent a significant challenge in musculoskeletal medicine, with limited therapeutic options due to the tissue’s poor healing capacity [4,8]. The need for effective non-invasive treatments has driven research toward novel therapeutic approaches, including various forms of bio-physical stimulation [21]. Our study provides the first comprehensive in vitro investigation on the effects of EMTT on human tenocytes, revealing several significant findings contributing to understanding its therapeutic potential. EMTT significantly influences multiple aspects of tenocyte behavior. The initial experiments with 40 mT exposure showed minimal effects on cell viability, while subsequent treatments at 80 mT induced more pronounced cellular responses. This dose-dependent effect aligns with previous studies on electromagnetic field therapies [19,33], suggesting the importance of optimizing treatment parameters. The observed enhancement of cell proliferation, evidenced by an increased Ki67 expression (+7% in treated cells), indicates that EMTT may stimulate the regenerative capacity of tendon tissue. Similar proliferative responses have been reported with ESWT [11,13], although our findings suggest that EMTT’s electromagnetic mechanism may offer distinct advantages regarding cellular response modulation.

A particularly intriguing finding is the dual effect observed with repeated 80 mT exposure: while inducing a decrease in overall cell vitality, it simultaneously promoted selective impact on the cell population. The significant downregulation of the senescence marker CDKN2a/INK4a, coupled with increased Ki67 expression, suggests potential senolytic-like effects. However, this interpretation requires careful consideration of alternative mechanisms. The observed pattern could also result from stress adaptation responses, where cells modulate cellular reprogramming or metabolic changes that affect the expression of cell cycle regulators independently of senescence status. The observed effects—overall reduction in cell viability but increased proliferation (Ki67+)—might suggest that EMTT selectively affects less responsive cells while simultaneously enabling healthy cells to enhance their proliferative activity. However, definitive proof of senolytic activity would require direct assessment through validated approaches, including senescence-associated β-galactosidase assays, measurement of senescence-associated secretory phenotype (SASP) factors, or single-cell RNA sequencing to characterize cellular heterogeneity and senescence states [34,35,36]. This preliminary evidence of potential senolytic-like effects distinguishes EMTT from current treatments such as PRP or stem cell therapy, which primarily focus on tissue regeneration without specifically addressing cellular senescence pathways [10,36]. Future studies should prioritize the mechanistic characterization of these effects to determine whether EMTT truly exhibits senolytic properties or if the observed changes reflect other cellular adaptation mechanisms. Recent research has highlighted the detrimental impact of senescent cells in tendon pathology [37,38], and their selective elimination could represent a novel therapeutic strategy.

Our analysis of cellular migration and cytoskeletal organization revealed striking effects of EMTT treatment. The enhanced migratory capacity, demonstrated by the scratch test results, was accompanied by significant changes in actin stress fiber formation and paxillin distribution. These modifications in cytoskeletal architecture are critical for proper tendon healing [39]. The increased expression of MMP9 in treated cells further supports the potential of EMTT to promote tissue remodeling, as matrix metalloproteinases play crucial roles in tendon repair and adaptation [27,28,40,41,42]. The molecular analysis of tenogenic markers provided particularly valuable insights. The upregulation of scleraxis (SCX) and tenomodulin (TNMD) in EMTT-treated cells suggests enhancement of the tenogenic phenotype. These findings are especially relevant, as the loss of tenogenic markers is a hallmark of tendinopathy [43]. The differential response observed with various treatment protocols (particularly evident in the 3500 impulses experiments) highlights the importance of optimizing treatment parameters for clinical applications. The experimental findings demonstrate substantial therapeutic potential in the regulation of collagen synthesis pathways, with implications for clinical applications. An increased production of Collagen I and III, followed in both preliminary and intensive treatment protocols, suggests that EMTT might promote appropriate matrix remodelling. This balance is crucial, as altered collagen ratios are associated with poor healing outcomes in tendinopathy [43]. Our results are consistent with recent studies demonstrating the importance of mechanical stimulation in regulating tendon extracellular matrix composition [42,43].

The strengths of our study include a systematic evaluation of multiple treatment parameters and a comprehensive analysis of cellular responses. Including early (Ki67, migration) and later (matrix production, differentiation) cellular responses, this study provides a broad understanding of EMTT’s biological effects. Nevertheless, several additional methodological limitations should be acknowledged. Our interpretation of potential senolytic effects, while supported by CDKN2a/INK4a downregulation, requires validation through more specific assays such as senescence-associated β-galactosidase staining or direct assessment of senescent cell clearance. Furthermore, our gene expression analysis, while comprehensive, represents only transcriptional changes, and protein-level validation would strengthen these findings. Additionally, flow cytometry analysis was not included in the original study design, which would have provided a more robust quantitative assessment of proliferation and senescence markers. Future investigations should incorporate these methodological approaches from the initial study design to provide a more definitive characterization of EMTT’s cellular effects.

The in vitro nature of our investigation, while providing detailed mechanistic insights, cannot fully replicate the complex in vivo environment of tendon tissue. Additionally, our use of healthy tenocytes might not wholly reflect the cellular responses in pathological conditions, where the cellular phenotype and matrix composition are already altered [42,43,44]. Our findings have several important clinical implications that could significantly impact the therapeutic approach to tendinopathies. Translating these laboratory findings to clinical practice suggests multiple promising applications for EMTT treatment. The demonstrated ability of EMTT to promote both proliferative and regenerative responses, while potentially eliminating senescent cells, indicates effectiveness in chronic tendinopathies, where tissue degeneration and cellular senescence are prominent features [43,44]. This dual action appears especially valuable in older patients or patients suffering from long-standing tendinopathy, where senescent cell accumulation impairs natural healing processes [38]. The enhanced migratory and matrix-production capabilities of EMTT-treated cells suggest potential applications in both post-surgical recovery and conservative management of partial tendon lesions. Our molecular findings, particularly the balanced upregulation of Collagen I-III and MMP expression modulation, indicate that EMTT might promote more physiological tissue repair rather than scarring. This aspect appears particularly relevant in treating partial tendon tears, where maintaining appropriate extracellular matrix composition is crucial for restoring mechanical properties [45].

Clinical applications of EMTT could be particularly beneficial in several scenarios. The proliferative and senolytic effects suggest efficacy as an early intervention in tendinopathies, potentially preventing disease progression. In post-surgical rehabilitation, enhanced cellular migration and matrix production could accelerate healing and reduce recovery time in patients with chronic pathology resistant to conventional treatments. The unique combination of regenerative and senolytic effects offers new therapeutic possibilities [15,16,17]. Additionally, athletes and workers with high mechanical demands on their tendons might benefit from prophylactic EMTT treatment to maintain tissue health [46]. Our findings regarding treatment parameters could guide clinical protocol development. The observed dose-dependent effects, particularly enhanced responses to repeated 80 mT exposures, provide a scientific basis for treatment intensity and frequency. However, the complex cellular responses observed with different protocols emphasize the need for personalized treatment approaches based on specific clinical conditions [47]. The molecular mechanisms described in the present study suggest potential synergies with current therapeutic strategies. The mechanobiological effects could complement and improve controlled exercise protocols [44], while potentially optimizing the local environment for transplanted stem cells, enhancing their survival and differentiation [44]. Practical considerations for clinical translation include optimal treatment timing, with our results suggesting benefits from frequent initial treatments followed by maintenance sessions. The observed effects on cell migration indicate the importance of treating primary lesion sites and surrounding areas. Furthermore, the identified molecular markers could serve as potential biomarkers for treatment response, though clinical correlation studies are needed [21]. Future research directions should address several key aspects of EMTT application. The development of standardized treatment protocols based on specific pathological conditions appears essential, alongside identifying reliable clinical outcome measures correlating with observed cellular effects. Long-term follow-up studies will be crucial for assessing treatment durability, while investigation of combination therapies could optimize outcomes [15]. The potential senolytic effects warrant further investigation through specific markers and functional assays, particularly for aging-related tendinopathies [17,18,19].

In conclusion, our study provides strong evidence that EMTT significantly influences tenocyte behavior through multiple mechanisms. The combination of proliferative, migratory, and matrix-modulating effects, along with preliminary evidence of potential senolytic-like activity, suggests EMTT as a promising treatment modality that warrants further clinical investigation. However, the mechanistic basis of the observed cellular responses, particularly regarding senescence pathways, requires additional validation through more specific methodological approaches. This work represents the first detailed characterization of the cellular and molecular effects of EMTT on human tenocytes, bridging an important knowledge gap between observed clinical efficacy and underlying biological mechanisms. Future research should focus on the mechanistic validation of these preliminary findings and their translation into practical clinical protocols, potentially revolutionizing the management of tendinopathies.

## 4. Materials and Methods

### 4.1. Cells and Treatments

Six primary cultures of human tenocytes were produced using samples derived from semitendinosus tendons harvested from patients who underwent arthroscopic ACL reconstruction (all males aged from 17 to 37 years), after informed consent was given, according to protocols approved by the Institutional Review Board of “San Giovanni di Dio e Ruggi D’Aragona Hospital” (Salerno, Italy) (Review Board prot./SCCE *n*. 151, 29 October 2020). All our samples were derived from the osteotendinous junction at the tibia. After being harvested, the tendons were cut into small pieces and digested with 2 mg/mL collagenase type I (GIBCO). The samples were centrifugated at 1000 rpm for 10 min; the supernatant was discarded, and the pellet was cultured in a sterile flask (Falcon), in D-MEM supplemented with 10% fetal bovine serum (FBS, Hyclone, Euroclone; Ridder Gerardlaan 50, 2650 Edegem, Belgium) and 1% penicillin/streptomycin/glutamine solution (GIBCO). Cells were incubated at 37 °C, 5% CO_2_. They were expanded to two passages (P2). Subsequently, when they reached a confluence of 80–85% at P3, they were used for the experiments.

The electromagnetic transduction treatment (EMTT) applied in the experiment derived from the use of an electromagnetic wave generator known as MAGNETOLITH (STORZ MEDICAL AG; Tagerwilen, Switzerland), whereas the control group was maintained in the same culture conditions, without EMTT treatment. Given the novelty of this treatment technique and, therefore, the lack of literature on the effects of this energy on in vitro cultures of tenocytes, we performed a preliminary experiment with a single exposure to evaluate its possible impact on cell vitality. After this, we exposed our samples to a more intense treatment, aiming to exacerbate cell response with more impulses, which were repeated in three different settings: one, two, or three consecutive applications. In the preliminary experiments, we exposed our cultured cells to 1000 electromagnetic wave impulses at two different intensities, 40 and 80 mT (to determine the intensity limit our cultured cells could cope with), with an impulse frequency of 5 Hz. After this evaluation, cultured cells were exposed to a more intense treatment of 3500 impulses, administered once, twice, or three times consecutively, receiving a total of 3500, 7000, and 10,500 impulses at an intensity of 80 mT with an impulse frequency of 5 Hz. Treated and untreated cells were incubated for 24 h and then collected for experiments.

### 4.2. Immunofluorescence

Immunofluorescence slides were prepared by placing them in a 24-well plate (Falcon) and covered with a thin layer of gelatin (2%) to allow cell adhesion as described [48]. Cell and growth medium were added to each well, and tenocytes were incubated for 24 h. Slides were then moved into a 3 cm dish and exposed to EMTT treatment. After 24 h, the cells were fixed in paraformaldehyde at 4%, treated with 0.1 M glycine and 0.1% Triton X-100 to allow membrane permeabilization. Since 30 min in blocking buffer(3% BSA), cells were alternatively incubated with the following primary antibodies: the rabbit polyclonal antibody anti-Ki67(1:100) (Invitrogen, Camarillo, CA, USA), mouse anti-paxillin monoclonal antibody (1:50) (BD Transduction Laboratories, San Diego, CA, USA), mouse monoclonal antibody anti-vimentin (1:50) (Dako, Denmark) mouse monoclonal antibody anti-Human pan-cytokeratin (1:50) (Dako, Denmark). TRITC–phalloidin was used to visualize cytoskeleton organization (Sigma Chemicals, St. Louis, MO, USA). Nuclear staining was performed by the addition of 1 µg/mL of DAPI (40,60-diamino-2-phenylindole) (1:10.000) (Sigma, Chemicals, St. Louis, MO, USA). Monoclonal and polyclonal antibodies were obtained from animals immunized with human antigens, which are recognized explicitly by those reagents. The primary antibodies were visualized using goat anti-mouse IgG-FITC (1:50; Cappel Research Products, Durham, NC, USA) and goat anti-rabbit IgG-Texas Red (1:200; Jackson Immunoresearch Laboratories, West Grove, PA, USA), goat anti-mouse IgG-Texas Red (1:200; Jackson Immunoresearch Laboratories, West Grove, PA, USA). All the signals were analyzed by recording and merging single-stained images with an Axiovert 200 inverted microscope, and image analysis was then performed using Axiovision software V 4.8.2 SP2 (Carl Zeiss MicroImaging GmbH, Oberkochen, Germany). For quantitative assessment of cell proliferation, Ki67-positive cells were counted using 10 fields (20×) randomly taken. ImageJ 1.54J software was used for fluorescence intensity evaluation to measure mean fluorescence intensity on 10 images for each treatment, with the same exposure time and randomly taken. Statistical analysis was performed using Student’s *t*-test.

### 4.3. Colorimetric MTT Assay

The 3-(4,5-dimethylthiazol)-2,5 diphenyl tetrazolium bromide (MTT; Sigma, Tokyo, Japan) assay was performed to measure mitochondrial cell activity after electromagnetic stimulation. Briefly, viable tenocytes were cultivated in dishes, and 24 h after treatment, cells were incubated for 3 h with MTT (1 mg/mL) and lysed in dimethyl sulfoxide. The absorbance at 570 nm was measured by a microplate reader (Multiscan Spectrum Thermo Electron Corporation, Vantaa, Finland).

### 4.4. Scratch Assay

Cells were seeded on 6 cm dishes, incubated for 24 h at 37 °C, then exposed to EMTT, and a standardized cell-free area was introduced by scraping the monolayer with a sterile tip [49]. Unexposed cells were used as controls. After intensive wash, the remaining cells were incubated for 24 h, fixed with 4% paraformaldehyde for 30 min at 25 °C, and photographs were taken using an Axiovert 200 inverted microscope (Zeiss). Some plates were fixed and photographed immediately after scratching, representing a T0 control. Migration was quantitated by measuring the recovered scratch area, which was performed using the Axiovision software (Zeiss). The data presented are the means of triplicate experiments ±SD.

### 4.5. RNA Extraction and cDNA Synthesis

RNA was extracted using the Quick-RNA MicroPrep Kit (Zymo Research, Irvine, CA, USA). Total RNA concentration was quantified by NanoDrop™ One/One c (Thermo Scientific, Waltham, MA, USA) 1 µg of total RNA was used for reverse transcription using the iScript™ cDNA synthesis kit (Bio-Rad, Hercules, CA, USA) according to the manufacturer’s instructions.

### 4.6. Primers

The oligonucleotide primers used to target genes, and the housekeeping gene were selected using the online tool Primer-BLAST purchased from Invitrogen [50]. We performed no-template control and no-reverse-transcriptase control (RT negative) assays for each primer pair, which produced negligible signals (Table 1).

### 4.7. RT-PCR Analysis

Real-time PCR was performed using the iCycler Real-Time Detection System (iQ5 Bio-Rad) with optimized PCR conditions [51]. The reaction was carried out in a 96-well plate using iQ SYBR Green Supermix (Bio-Rad), adding forward and reverse primers for each gene and 1 µL of diluted template cDNA to a final reaction volume of 15 µL. All assays included a negative control and were replicated three times. The thermal cycling program was performed as follows: an initial denaturation step at 95 °C for 3 min, followed by 45 cycles at 95 °C for 10 sec. and 60 °C for 30 sec. According to the manufacturer’s manual, real-time quantitation was performed with the help of the iCycler IQ optical system software version 3.0 a (Bio-Rad). The relative expression of the housekeeping gene was used to standardize the reaction. The comparative threshold cycle (Ct) method was applied to calculate the fold changes in expression compared to control cells. The results are reported as mean standard deviation (SD) from three experiments in triplicate.

### 4.8. Statistical Analysis

Data was analyzed using GraphPad Prism software version 10.0 0 (GraphPad Software, La Jolla, CA, USA). MTT assay, ki67 immunofluorescence quantification, and real-time RT-PCR data were evaluated using ordinary one-way analysis of variance (ANOVA) followed by multiple comparison tests to assess differences among all groups and between selected pairs. Unpaired Student’s *t*-test was used to analyze scratch assay results, as well as for fluorescence intensity evaluation. A *p* value < 0.05 was considered statistically significant. Data are presented as the mean ± standard deviation (SD) from three independent experiments, each performed in triplicate.

## Figures and Tables

**Figure 1 ijms-26-07122-f001:**
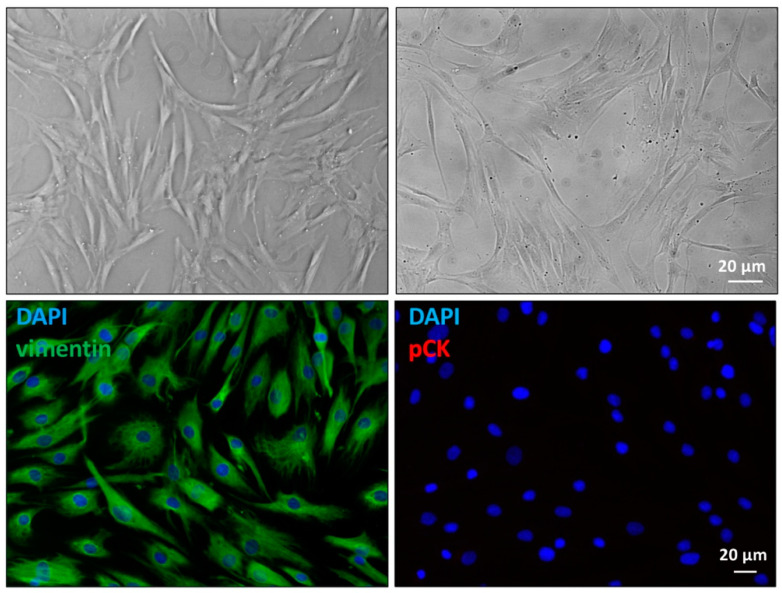
Images of the primary culture of human tenocytes. Top panels, differential interference contrast (**left**) and phase contrast microscopy (**right**). Bottom panels, cells that were stained with 4′,6-diamidino-2-phenylindole (DAPI) and vimentin (**left**), or human pan-cytokeratin (pCK) (**right**).

**Figure 2 ijms-26-07122-f002:**
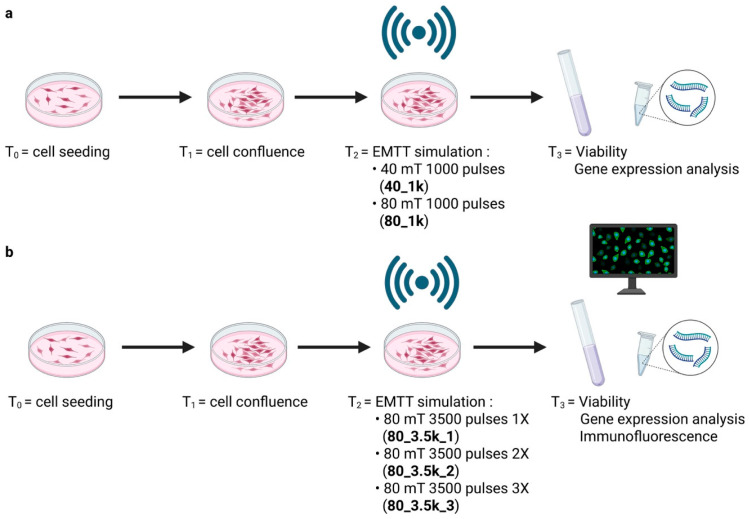
Experimental design of the dose–response study (T1 = 48 h; T2 = 54 h; T3 = 55 h). (**a**) Two field intensities approach. (**b**) Higher field intensities with variable number of pulses approach. Created with https://BioRender.com on 27 April 2025.

**Figure 3 ijms-26-07122-f003:**
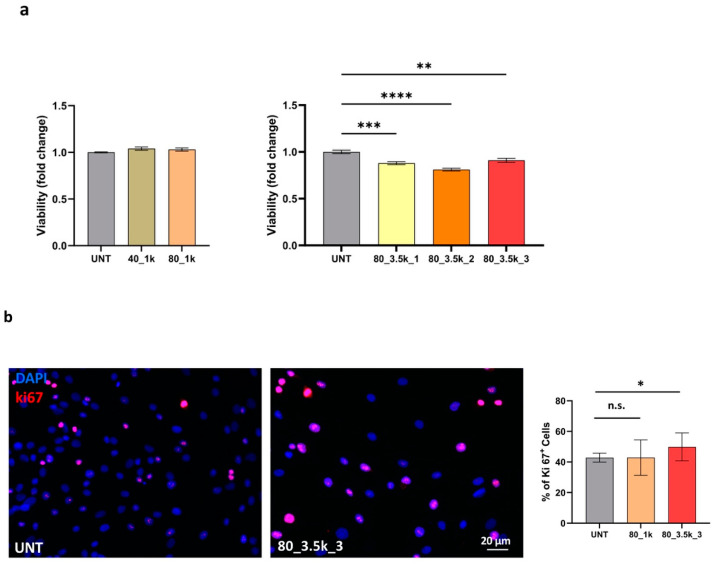
Cell viability and proliferation evaluation in EMTT-treated cells. (**a**) Cells treated as described above were evaluated for vitality by colorimetric MTT assay. (**b**) Cells treated as above were stained for immunofluorescence, and nuclei were visualized by DAPI staining. Bar: The proliferation rate, assessed by immunofluorescence with an anti-Ki67 polyclonal antibody, reveals positive cycling cells comparable in all the experimental conditions; 20 µm. The results are expressed as mean fold increase ± standard deviation (SD), * *p* < 0.05, ** *p* < 0.01, *** *p* < 0.001, **** *p* < 0.0001. n.s. means not significant.

**Figure 4 ijms-26-07122-f004:**
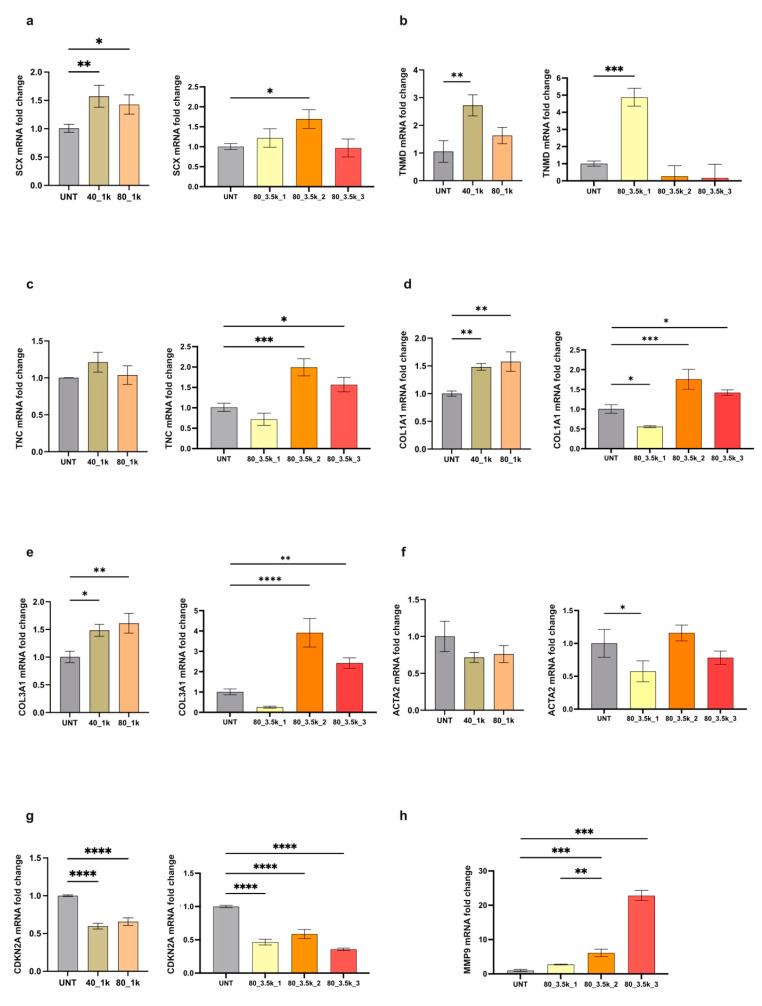
Modulation of mRNA expression levels of differentiation markers after EMTT exposure. (**a**–**h**) Bar graphs showing gene expression levels of the different markers: scleraxis, SCX (**a**); tenomodulin, TNMD (**b**); tenascin-C, TNC (**c**); type I collagen, COL1A1 (**d**); type III collagen, COL3A1 (**e**); alpha smooth muscle actin (α-SMA), ACTA2 (**f**); cyclin-dependent kinase inhibitor 2A (p16), CDKN2A (**g**); matrix metalloproteinase-9, MMP9 (**h**) in cells treated as above compared to untreated cells (UNT). Results are expressed as mean value ± SD. ns, not statistically significant; * *p* < 0.05; ** *p* < 0.01; *** *p* < 0.001, **** *p* < 0.0001.

**Figure 5 ijms-26-07122-f005:**
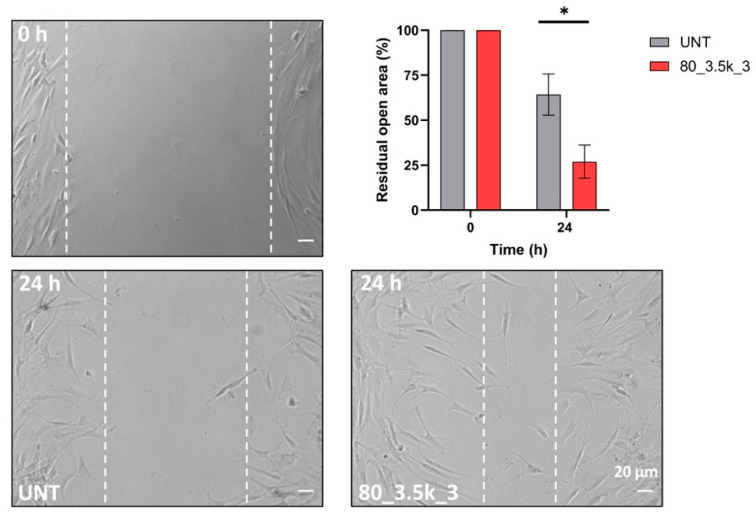
Effects of EMTT exposure on tenocytes’ migration abilities. Representative images of scratch assay (*n* = 3) showing the effect of EMTT on cell migration. Bar: 20 µm. The percentage of residual open area after 24 h of treatment with EMTT at 80_3.5k_3, compared to that of untreated cells (UNT), was measured using the Axiovision software as reported in the Materials and Methods section. Results are expressed as mean value ± SD, * *p* < 0.05.

**Figure 6 ijms-26-07122-f006:**
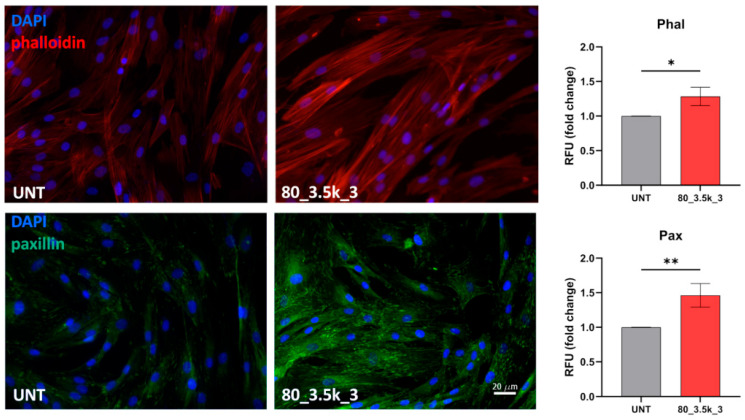
Immunofluorescence analysis of the effects of acute EMTT exposure on actin cytoskeleton architecture and paxillin expression in tenocytes. Representative IF acquisitions (*n* = 3) showing actin cytoskeleton (top panels) and paxillin expression (bottom panels) of cells treated with EMTT at 80_3.5k_3 or untreated (UNT). F-Actin is stained in red with TRITC–Phalloidin, paxillin is stained in green with FITC-conjugated antibody, and nuclei are stained in blue with DAPI. Bar: 10 µm. Quantitative immunofluorescence analysis of TRITC–Phalloidin (top graph) and FITC–paxillin (bottom graph) staining is expressed as the mean ± SD relative fluorescence unit normalized to untreated cells (* *p* < 0.05; ** *p* < 0.01).

**Table 1 ijms-26-07122-t001:** Primers used for targets and housekeeping genes.

Gene	Primers
*Col I*	5′–ACATGTTCAGCTTTGTGGACCTCCG-3′/5′–ACGCAGGTGATTGGTGGGAGTTCT-3′
*Col III*	5′–AGGGTGTCAAGGGTGAAAGTGGGA-3′/5′–ACCAGCCAGACCAGGAAGACCC-3′
*Scx*	5′–CGAGCGAGACCGCACCAACA-3′/5′–CGTTGCCCAGGTGCGAGATGTAG-3′
*Tnm*	5′–CCGCCGCGTCTGTGAACCTT-3′/5′–GCGGGCCACCCACCAGTTAC-3′
*Tn-C*	5′–GGAGGGGACCACGCTAGGT-3′/5′–TCCCGGCCTAGACCTGTGAG-3′
*18 s*	5′–CGAGCCGCCTGGATACC-3′/5′–CATGGCCTCAGTTCCGAAAA-3′
*CDKN2A*	5′–CGTGGACCTGGCTGAGGA-3′/5′–AATCGGGGATGTCTGAGGGA-3′
*Alpha-SMA*	5′–GCACCCCTAGAACCCCAAG-3′/5′–ACGATGCCAGTTGTGCGT-3′
*MMP9*	5′–CGCGCTGGGCTTAGATCATT-3ʹ/5′–GGGCGAGGACCATAGAGGT-3ʹ

## Data Availability

All the data are provided in the article.

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
