# Peer review of "Electromagnetic Transduction Therapy (EMTT) Enhances Tenocyte Regenerative Potential: Evidence for Senolytic-like Effects and Matrix Remodeling"

_ijms, 2025, doi:10.3390/ijms26157122_

Round 1

Reviewer 1 Report

Comments and Suggestions for Authors

The manuscript presents an original and valuable in vitro investigation on the biological effects of Electromagnetic Transduction Therapy (EMTT) on primary human tenocytes. It explores several key cellular processes, including proliferation, migration, senescence, cytoskeletal remodeling, and matrix-related gene expression. The topic is both timely and relevant, given the increasing application of EMTT in musculoskeletal medicine, despite the current lack of mechanistic understanding of its effects. The study is well-structured, comprehensive, and supported by an adequate methodological approach. The discussion is thoughtful and well contextualized within the existing literature. Overall, the manuscript appears suitable for publication in IJMS following the resolution of several important revisions and clarifications.

One area that requires particular attention is the interpretation of the so-called senolytic-like effect, based on the downregulation of CDKN2a/INK4a and selective cell loss. While the findings are intriguing, this interpretation currently remains somewhat speculative. The authors should strengthen their argument by discussing potential alternative mechanisms, such as stress adaptation rather than selective clearance of senescent cells, and by highlighting the need for future studies using validated approaches, including senescence-associated β-galactosidase assays or single-cell analyses.

In addition, the description of the statistical analysis is insufficient and should be considerably expanded. The manuscript does not provide clear information regarding the types of tests employed (for example, ANOVA, t-test, or non-parametric tests), the use of corrections for multiple comparisons, the software used, or whether assumptions of normality and homogeneity of variance were assessed. This omission is concerning, especially in light of certain inconsistencies observed in the figures. Several untreated control (UNT) groups are presented without error bars, which impedes visual assessment of data variability and statistical significance. Conversely, some treated groups exhibit very wide error bars, suggesting either high data variability or the possibility of a non-parametric data distribution. Furthermore, it remains unclear whether the same number of replicates (n) was used consistently across different experimental conditions, and this should be explicitly clarified.

Author Response

General Response of the authors to Comments: 

We thank the reviewer for their valuable comments on our work, which have helped us improve the manuscript. Based on these suggestions, we have made some revisions in an effort to address each point raised.

Point-by-point response to Comments:

Comment 1: One area that requires particular attention is the interpretation of the so-called senolytic-like effect, based on the downregulation of CDKN2a/INK4a and selective cell loss. While the findings are intriguing, this interpretation currently remains somewhat speculative. The authors should strengthen their argument by discussing potential alternative mechanisms, such as stress adaptation rather than selective clearance of senescent cells, and by highlighting the need for future studies using validated approaches, including senescence-associated β-galactosidase assays or single-cell analyses.

Response 1: We thank the reviewer for this observation regarding our interpretation of the senolytic-like effects. We acknowledge that our interpretation requires more detailed discussion and have revised the manuscript accordingly. We have modified the text to:

Clarify the terminology: We now refer to "senolytic-like effects" rather than definitive senolytic activity to acknowledge the preliminary nature of our findings.

Discuss alternative mechanisms: We have added a discussion of stress adaptation responses and other potential mechanisms that could explain the observed CDKN2a/INK4a downregulation.

Highlight study limitations: We explicitly acknowledge that definitive proof of senolytic activity would require additional validated approaches such as senescence-associated β-galactosidase assays, single-cell RNA sequencing, or direct assessment of senescent cell clearance.

Moderate conclusions: We have tempered our conclusions to present these findings as preliminary evidence requiring further validation.

The revised discussion now presents our findings as suggestive evidence that warrants further investigation rather than definitive proof of senolytic activity, which better reflects the scope of our current methodology.

In particular:

Title's modifications: "Electromagnetic Transduction Therapy (EMTT) enhances tenocyte regenerative potential: evidence for senolytic-like effects and matrix remodelling"

Abstract's modifications: 

a) line.....: significant downregulation of senescence marker CDKN2a/INK4a were observed, suggesting potential senolytic-like effects

b) line.....: while simultaneously exhibiting potential senolytic-like effects through downregulation of senescence-associated markers

Result Section's modification:

line.....: Taken together, our results indicate that human primary cultured tendon-derived cells exposed to EMTT are affected by the treatments in different ways. This is demonstrated by variations in collagen production, metalloproteinase expression, and changes in tenocyte differentiation markers, which are crucial in the mechanisms guiding injury repair. The observed downregulation of CDKN2a suggests potential effects on senescence pathways, though definitive characterization of senolytic activity would require additional validated approaches such as senescence-associated β-galactosidase assays or single-cell analyses 

Discussion Section's modification:

line.....: A particularly intriguing finding is the dual effect observed with repeated 80 mT exposure: while inducing a decrease in overall cell vitality, it simultaneously promoted selective impact on the cell population. The significant downregulation of the senescence marker CDKN2a/INK4a, coupled with increased Ki67 expression, suggests potential senolytic-like effects. However, this interpretation requires careful consideration of alternative mechanisms. The observed pattern could also result from stress adaptation responses, where cells modulate senescence-associated gene expression without necessarily involving selective elimination of senescent cells. Additionally, the electromagnetic stimulation might induce cellular reprogramming or metabolic changes that affect the expression of cell cycle regulators independently of senescence status. Recent research has highlighted the detrimental impact of senescent cells in tendon pathology, and their selective elimination could represent a novel therapeutic strategy. The observed effects —overall reduction in cell viability but increased proliferation (Ki67+)— might suggest that EMTT selectively affects less responsive cells while simultaneously enabling healthy cells to enhance their proliferative activity. However, definitive proof of senolytic activity would require direct assessment through validated approaches including senescence-associated β-galactosidase assays, measurement of senescence-associated secretory phenotype (SASP) factors, or single-cell RNA sequencing to characterize cellular heterogeneity and senescence states . This preliminary evidence of potential senolytic-like effects distinguishes EMTT from current treatments such as PRP or stem cell therapy, which primarily focus on tissue regeneration without specifically addressing cellular senescence pathways. Future studies should prioritize the mechanistic characterization of these effects to determine whether EMTT truly exhibits senolytic properties or if the observed changes reflect other cellular adaptation mechanisms.

Limitations Section's modification:

Several additional methodological limitations should be acknowledged. Our interpretation of potential senolytic effects, while supported by CDKN2a/INK4a downregulation, requires validation through more specific assays such as senescence-associated β-galactosidase staining or direct assessment of senescent cell clearance. Furthermore, our gene expression analysis, while comprehensive, represents only transcriptional changes, and protein-level validation would strengthen these findings. Additionally, flow cytometry analysis was not included in the original study design, which would have provided more robust quantitative assessment of proliferation and senescence markers. Future investigations should incorporate these methodological approaches from the initial study design to provide more definitive characterization of EMTT's cellular effects.

Conclusions modification:

In conclusion, our study provides strong evidence that EMTT significantly influences tenocyte behavior through multiple mechanisms. The combination of proliferative, migratory, and matrix-modulating effects, along with preliminary evidence of potential senolytic-like activity, suggests EMTT as a promising treatment modality that warrants further clinical investigation. However, the mechanistic basis of the observed cellular responses, particularly regarding senescence pathways, requires additional validation through more specific methodological approaches. This work represents the first detailed characterization of the cellular and molecular effects of EMTT on human tenocytes, bridging an important knowledge gap between observed clinical efficacy and underlying biological mechanisms. Future research should focus on mechanistic validation of these preliminary findings and their translation into practical clinical protocols, potentially revolutionizing the management of tendinopathies.

Comment 2: In addition, the description of the statistical analysis is insufficient and should be considerably expanded. The manuscript does not provide clear information regarding the types of tests employed (for example, ANOVA, t-test, or non-parametric tests), the use of corrections for multiple comparisons, the software used, or whether assumptions of normality and homogeneity of variance were assessed. This omission is concerning, especially in light of certain inconsistencies observed in the figures. Several untreated control (UNT) groups are presented without error bars, which impedes visual assessment of data variability and statistical significance. Conversely, some treated groups exhibit very wide error bars, suggesting either high data variability or the possibility of a non-parametric data distribution. Furthermore, it remains unclear whether the same number of replicates (n) was used consistently across different experimental conditions, and this should be explicitly clarified.

Response 2: We thank the reviewer for highlighting these important issues, which have helped us significantly improve the quality of our manuscript. In response, we have added a new paragraph to the Materials and Methods section, where the statistical approach is now described in detail, including the types of tests used, software, assumptions checked, and corrections applied for multiple comparisons. We have also carefully reviewed the figures and identified instances where error bars for control (UNT) groups were incorrectly omitted. These have been corrected. In some cases, the standard deviation (SD) was so small that the error bars were not visible; this occurred when the three replicates yielded nearly identical values. We emphasize, as written in the new paragraph of the statistics, that all experiments were performed in triplicate (n = 3), and that the data presented in the figures represent the mean ± SD of these biological replicates. 

This new paragraph was added:

4.8. Statistical analysis

Data were analyzed using GraphPad Prism software version 10.0 0 (GraphPad Software, La Jolla, USA). MTT assay, ki67 immunofluorescence quantification and real time RT-PCR data were evaluated using one-way analysis of variance (ANOVA) followed by multiple comparison tests to assess differences among all groups and between selected pairs. Unpaired student’s t-test was used to analyze scratch assay results as well as for fluorescence intensity evaluation. A p value < 0.05 was considered statistically significant. Data are presented as the mean ± standard deviation (SD)  from three independent experiments, each performed in triplicate.

Reviewer 2 Report

Comments and Suggestions for Authors

In this paper, Mancini et al. studied the effect of Electromagnetic Transduction Therapy (EMTT) on human isolated tenocyte proliferation, senescence, and migration. The authors showed that EMTT treatment increases tenocyte proliferation and migration. They also found an increase in tenogenic markers and collagen I and III expression upon EMTT treatment, suggesting a potential implication in the treatment of tendinopathies. The authors’ approach is novel and may be useful in tendinopathy management. I have a few major comments for further improvement of the manuscript:

  • The authors should provide a brief introduction about the working principle of EMTT in the Introduction section.
  • Figure 3b: The magnification of untreated and treated images does not appear to be the same. Please confirm with the raw data.
  • The authors need to corroborate their cell proliferation and senescence data using flow cytometry.
  • For the various expression studies, the authors only showed the qRT-PCR data. They need to confirm their findings by western blotting, as mRNA expression may not necessarily always be associated with changes in protein expression.
  • For FITC-paxillin staining, try to reduce the background by changing the blocking buffer and, if possible, image them at 60X magnification. In the current images, the paxillin spots are not very clear due to background and magnification issues.

Author Response

General Response of the authors to Comments: 

We thank the reviewer for their valuable comments and suggestions, which have been taken into account to enhance the quality of our research work.

Point-by-point response to Comments:

Comment 1: The authors should provide a brief introduction about the working principle of EMTT in the Introduction section.

Response 1: The following sentence has been added to the text in the Introduction's section. "Electromagnetic Transduction Therapy (EMTT) is a non-invasive regenerative modality that delivers high-frequency pulsed electromagnetic fields to stimulate cellular activity and promote tissue repair".

Comment 2: The authors need to corroborate their cell proliferation and senescence data using flow cytometry. For the various expression studies, the authors only showed the qRT-PCR data. They need to confirm their findings by western blotting, as mRNA expression may not necessarily always be associated with changes in protein expression.

Response 2: We appreciate the reviewer's suggestions for additional methodological approaches to strengthen our findings. We agree that flow cytometry analysis would provide valuable quantitative confirmation of our proliferation and senescence findings. However, flow cytometry analysis was not included in the original study design and protocol approval, and is therefore not feasible within the current study framework. We recognize this as a limitation of our study and have added this to the limitations section. Future studies investigating EMTT effects on tenocytes should incorporate flow cytometry analysis from the initial study design to provide a more robust quantitative assessment of cellular responses. 

Nevertheless, we also agree with the reviewer that protein-level confirmation of our gene expression findings would significantly strengthen the manuscript. Nonetheless, we have scheduled Western blot analyses targeting key proteins -scleraxis (SCX), tenomodulin (TNMD), and collagen I (COL1A1)- to corroborate our qRT-PCR findings and ensure a thorough molecular assessment at both the gene and protein expression levels. However, the journal editor explicitly requested that, according to the reviewers' comments, we upload the revised files within 10 days, a time frame that was too limited to complete the requested experiments. We have also added a statement in the limitations section acknowledging that mRNA expression levels do not always correlate directly with protein expression, and that future studies should incorporate both transcriptional and proteomic approaches for complete molecular characterization.

Comment 3: Figure 3b: The magnification of the untreated and treated images does not appear to be the same. Please confirm with the raw data.

Response 3: Regarding the observation on Figure 3b, although the images may seem to be at different magnifications, we have checked the raw data of the images in our possession, and we can state that the 20x magnification is the same for both images. We attach the images with the related metadata files in the supplementary materials.

Comment 4: For FITC-paxillin staining, try to reduce the background by changing the blocking buffer and, if possible, image them at 60X magnification. In the current images, the paxillin spots are not very clear due to background and magnification issues.

Response 4: We are grateful to the reviewer for the valuable suggestion regarding FITC-paxillin staining. In response, we repeated the immunofluorescence procedure, enhancing labeling specificity by employing a more effective blocking buffer. Although we currently lack an objective lens that allows for higher magnification, we have included in the manuscript new representative images obtained from the updated immunofluorescence experiment.

Round 2

Reviewer 1 Report

Comments and Suggestions for Authors

The quality of some figures is unfortunately unreadable or too low to interpret properly. Kindly consider enhancing their resolution and clarity.

Author Response

Thank you for your valuable comments. I fully agree that some figures appear to be of low resolution and therefore difficult to interpret. I believe the issue stemmed from the editing process within the Word template required by the journal for manuscript submission. Therefore, the image quality has now been significantly improved, and its resolution has been appropriately enhanced.

Reviewer 2 Report

Comments and Suggestions for Authors

The authors have provided satisfactory responses to all the comments. Some of the suggested experiments they could not perform due to technical limitations, but they have included a statement in the discussion section. I do not have any further comments/suggestions. 

Author Response

Thanks for the comments.
